# IL-10 and IL-1β Serum Levels, Genetic Variants, and Metabolic Syndrome: Insights into Older Adults’ Clinical Characteristics

**DOI:** 10.3390/nu16081241

**Published:** 2024-04-22

**Authors:** Renata de Souza Freitas, Calliandra Maria de Souza Silva, Caroline Ferreira Fratelli, Luciano Ramos de Lima, Marina Morato Stival, Silvana Schwerz Funghetto, Izabel Cristina Rodrigues da Silva, Rosângela Vieira de Andrade

**Affiliations:** 1Graduate Program in Genomic Sciences and Biotechnology, Catholic University of Brasilia, Federal District, Brasília 72220-900, Brazilrosangelav@p.ucb.br (R.V.d.A.); 2Graduate Program in Health Sciences and Technologies, Faculty of Ceilândia, University of Brasília, Federal District, Brasília 72220-900, Brazil; 3Faculty of Ceilândia, University of Brasilia, Federal District, Brasília 72220-900, Brazil

**Keywords:** metabolic syndrome, interlekine-10, interlekine-1β, older population, *IL10* rs1800890, *IL1B* rs1143627

## Abstract

Populational aging is marked by chronic noncommunicable diseases, such as metabolic syndrome (MetS). IL-10 and IL-1β are pleiotropic cytokines with multiple biological effects linked to metabolic disorders. This cross-sectional study assessed 193 participants’ IL-10 and IL-1β serum levels regarding their role in developing MetS, clinical characteristics, and their *IL1B* rs1143627 and *IL10* rs1800890 variants’ genotype frequencies in a population over 60. IL-10 levels correlated weakly with HDL levels and fat mass and inversely with triglycerides, glucose, glycated hemoglobin, and estimated average blood glucose levels. IL-10 levels were also indirectly influenced by the patient’s T2DM duration, lean mass amount, and bone mineral content. Participants with altered HDL, elevated serum glucose, raised HbA1c levels, or those over 80 had reduced serum IL-10 levels compared to those with normal levels or other age groups, respectively. Women also had higher serum IL-10 levels than men. Dissimilarly, IL-1β levels correlated directly only with the number of total leukocytes and segmented neutrophils, showing only significant variations with self-reported alcohol consumption. Our study also found that those with the *IL10* AA genotype (lower IL-10 levels) had a significantly higher risk of developing MetS. These findings may help direct future research and more targeted therapeutic approaches in older adults.

## 1. Introduction

Population aging, or demographic transition, occurs due to a simultaneous decline in mortality and in the birth rate or fertility. The Brazilian Institute of Geography and Statistics (IBGE) predicts that, in 2030, Brazil will have around 40 million people aged 60 or over, surpassing the total number of children between zero and 14 years old [1,2]. This demographic transition leads to an epidemiological transition (a long-term shift in mortality and disease [3]) marked by a loss of physiological reserves and an emergence of chronic noncommunicable diseases (NCDs) that commonly come with the aging process. Consequently, aging is a risk factor for NCDs, such as metabolic syndrome (MetS), in which increased age is related to higher MetS prevalence [2,4,5].

MetS, whose base is insulin resistance, comprises a set of metabolic dysregulations, neurohormonal activations, and risk factors that manifest in an individual, increasing their risk of developing cardiovascular diseases (CVDs) and other NCDs, such as type 2 diabetes mellitus (T2DM), stroke/cerebrovascular accident (CVA), acute myocardial infarction (AMI), and even, eventually, neurological diseases [5,6,7,8,9]. Its clinical diagnosis can be defined by the presence of three or more criteria from the National Cholesterol Education Program’s Adult Treatment Panel III (NCEP-ATP III), which include increased abdominal (central) obesity, hypertension, elevated fasting blood glucose levels (hyperglycemia/dysglycemia), and atherogenic dyslipidemia, such as elevated serum triglyceride levels and reduced serum high-density lipoproteins (HDL) [5,6,7].

MetS is a low-grade (sub-clinical) chronic inflammatory disease whose pathogenesis encompasses multiple genetic, epigenetic, and acquired commodities interacting through several complex mechanisms yet to be fully elucidated [6,8]. If left untreated, the progressive increase in insulin-resistant adipose tissue, particularly visceral fat, impairs insulin-mediated inhibition of lipolysis and, thus, increases circulating free fatty acids (FFAs), which promotes gluconeogenesis and lipogenesis in the liver that ultimately result in atherogenic dyslipidemia [6,8]. Insulin resistance can also reduce the blood supply to adipocytes with consequent hypoxia partly due to the loss of insulin’s vasodilatory effect, FFA-induced vasoconstriction, and higher serum viscosity, creating a prothrombotic state that, together with increased leptins levels, increases inflammatory cytokine release from the adipose tissue [6,10].

Interleukins mediate and regulate inflammatory reactions and may differ in their immunological function. Among them, two stand out: interleukin IL-1β, a proinflammatory cytokine, and interleukin IL-10, an anti-inflammatory cytokine, both pleiotropic cytokines with multiple biological effects [6,10,11,12,13]. IL-10 is linked to metabolism as it can maintain insulin sensitivity, reduce glucose intolerance, increase whole-body lipid synthesis and skeletal muscle glycolysis, and decrease intramuscular fatty acyl-CoA levels [13,14,15,16,17,18,19,20]. Furthermore, alterations in cellular metabolism and specific metabolites, such as FFAs from the adipose tissue, regulate IL-10 production in different immune cells [18,21]. Likewise, IL-1β plays a critical role in physiological and pathological metabolism modulation (energy homeostasis), including insulin action and secretion, β-cell apoptosis, lipid metabolism, and food intake, by regulating the immune system and the neuronal and endocrine systems’ interface and, hence, in metabolic disorders, such as T2DM, MetS, and obesity [13,22,23,24,25,26].

Functional polymorphisms can also modify their genes’ expression and production, thus affecting their function. Both *IL10* and *IL1B* cytokine genes are highly polymorphic [27,28]. The *IL1B* rs1143627 (-31T>C) single nucleotide polymorphism (SNP) located in the gene’s promoter region, for instance, disrupts a TATA-box motif, markedly affecting several transcription factors’ binding affinity and thereby impacting *IL1B* transcription activity [27]. Another example, also located in its gene’s promoter region, is the *IL10* rs1800890 (-3575T>A) SNP polymorphism that affects IL-10 production levels [13,28,29]. Both polymorphisms could either protect or predispose an individual to MetS.

As various pathogenic pathways contributing to MetS development culminate in an inflammatory state, this study assessed the serum levels of IL-10 and *IL-1β* cytokines regarding their role in the susceptibility to developing MetS and other clinical characteristics in an older population (over 60). The study also compared the genotype frequencies of *IL1B* rs1143627 (-31T>C) and *IL10* rs1800890 (-3575T>A) variants in these research participants to their respective serum concentrations.

## 2. Materials and Methods

### 2.1. Research Participants

This study employed a cross-sectional design with both quantitative and qualitative approaches to observe, analyze, and describe patients treated at a basic health unit (UBS) in the Ceilândia Administrative Region of the Federal District—Brazil. By 2016, this region had the lowest family health coverage of the DF, with about 22% [30]. The sample consisted of older adults who voluntarily agreed to participate in the study after being invited. All participants were registered and received care from the Family Health Strategy teams at UBS No. 6 in the Ceilândia Region. A total of 193 participants were selected, taking into account a 5% margin of error and a 95% confidence level.

The inclusion criteria required participants to be 60 or older, registered with primary care services, monitored in the pre-selected unit, and capable of understanding, verbalizing, and answering the proposed questions. Concurrently, the study excluded participants if they were diagnosed with mental or psychiatric illness, were undergoing cancer treatment, had cardiac (heart) surgery within the past six months, had a pacemaker or metal prosthesis, were taking hormone replacement therapy, or were taking vitamins or supplements that could interfere with the measurement of biochemical or cytokine blood levels.

Recruited elders signed the informed consent form (ICF) after being fully informed about the stages of data collection. A nursing appointment was scheduled, during which the identification form was filled out with data such as weight, height, and health and socioeconomic factors. Participants also signed the Biological Material Custody Agreement, and their venous blood was collected.

This study was approved by the Research Ethics Committee (CEP) of the Health Sciences Teaching and Research Foundation (FEPECS) of the Federal District State Health Department (SES-DF) under opinion number 1.989.964/29 March 2017 with CAAE 59071116.8.0000.0030.

### 2.2. Sample Collection

Samples were collected from February to June 2019. The data collection process occurred during the nursing appointment at the UBS and consisted of two stages: blood collection and data collection. During the first stage, participants fasted for 8 to 12 h before providing approximately 15 mL of venous blood from the cubital fossa region. Subsequently, they received breakfast.

In the second stage, a structured data collection questionnaire was applied to evaluate the participants’ demographic and socioeconomic profiles. The questionnaire consisted of closed questions regarding biological sex, age, education, retirement, income, lifestyle habits (smoking), clinical conditions (hypertension or diabetes), diagnosis of T2DM, and use of medications (type and route of administration).

### 2.3. Metabolic Syndrome Defining Condition

The Brazilian Cardiological Society (BCS) criteria used to define metabolic syndrome (MetS) include waist circumference (WC) >88 cm (female) and >102 cm (male), blood pressure (BP) ≥130 × 85 mmHg (≥130/85) or use of antihypertensive medication, fasting blood glucose ≥110 mg/dL or use of T2DM medications, triglycerides (TG) ≥150 mg/dL or use of medications for dyslipidemia, and HDL <50 mg/dL (female) and <40 mg/dL (male) and require for a MetS diagnosis the presence of at least three criteria. These criteria are based on the National Cholesterol Evaluation Program for Adult Treatment Panel III (NCEP-ATP III) and the Brazilian Guideline for Diagnosis and Treatment of Metabolic Syndrome, which does not require insulin measurement but allows for it to be used as a clinical criterion [5].

### 2.4. Laboratory Analysis and Cytokine Dosage

Biochemical and hematological tests were performed at a Federal District’s private clinical analysis laboratory funded by the research project budget. These exams included the analysis of total leukocytes, rod neutrophils, segmented neutrophils, eosinophils, basophils, lymphocytes, and monocytes and measurements of total cholesterol, triglycerides, HDL, LDL, total lipids, glucose, glycated hemoglobin (HbA1c), estimated mean blood glucose, glutamic oxaloacetic transaminase, and glutamic pyruvic transaminase.

IL-10 and IL-1β levels were measured in our university’s clinical laboratory. These measurements were performed in triplicate using the enzyme-linked immunosorbent assay (ELISA) method according to the specifications of the R&D Systems Quantikine^®^ High-Sensitivity ELISA kit (Minneapolis, MN, USA). The samples’ mean values were calculated from these triplicate determinations. The sensitivity and the intra-assay coefficient of variation (CV), i.e., repeatability, for these ELISA were also determined. The control triplicate’s average determined the cutoff point, thus assuming a value of 2.0 pg/mL for IL-1β and 1.0 pg/mL for IL-10.

### 2.5. DNA Extraction

DNA was extracted from peripheral blood using the PureLink^®^ Genomic DNA Mini Kit from Invitrogen (catalog #K1820-02, lot #19339891 (Invitrogen Life Technologies, Carlsbad, CA, USA) following the manufacturer’s protocol, with an average yield of 20 ng/µL. The DNA was then stored at −70 °C until further use.

### 2.6. Genotyping

After DNA extraction, a diluted sample underwent a polymorphism polymerase chain reaction (PCR) using the Techne Thermal Cycler model TC-512 (Minneapolis, MN, USA) to study the distribution of single nucleotide polymorphisms (SNPs). The oligonucleotide sequences used to assess the polymorphisms were sense 5′—TAGTCCCCTCCCCTAAGAAGC—3′ and antisense 5′—CCCAGAATATTTCCCGAGTCA—3′ for *IL1B* −31T>C (rs1143627) SNP and sense 5′—GGTTTTCCTTCATTTGCAGC—3′ and antisense 5′—ACACTGTGAGCTTCTTGAGG—3′ for *IL10* -3575T>A (rs1800890) SNP.

Thermocycling conditions for the *IL1B* rs1143627 polymorphism were as follows: initial denaturation at 96 °C for 1 min, followed by 35 cycles of denaturation at 94 °C for 50 s, annealing of the oligonucleotides at 50 °C for 1 min, and extension of the fragments at 72 °C for 40 s for the extension of the fragments. The final extension was performed at 72 °C for 10 min and cooled at 4 °C for 60 min. Each reaction used 4.0 µL of genomic DNA at a final concentration of 2.5 ng/µL, 2.5 µL of 10× buffer (10 mM Tris and 50 mM KCl), 0.5 µL of 50 mM MgCl_2_ (Ludwig Biotec, Alvorada, Rio Grande do Sul, Brazil), 0.5 µL of 2.5 mM deoxyribonucleotide triphosphates (dNTPs) (Ludwig Biotec, Alvorada, Rio Grande do Sul, Brazil), 0.5 µL of 5 U/µL Taq-Polymerase (Ludwig Biotec, Alvorada, Rio Grande do Sul, Brazil), and 1.5 µL of each forward and reverse oligonucleotide (10 µM, IDT Technologies), completing with ultrapure water to a final volume of 25 µL per reaction.

The *IL1B* PCR product, a 421 bp fragment, was digested with the *Ava*I restriction enzyme (New England Biolabs, Inc. Beverly, MA, USA). Allele 1 (C) creates two restriction sites, resulting in three cleavage fragments, 344, 57 bp, and 20 bp, while allele 2 (T) is cleaved into four fragments (three restriction sites): 247, 97, 57 and 20 bp. This digestion system was assembled using 10.0 µL of PCR product, 2.0 µL of 10× NEB4 buffer (Biolabs), and 1 µL of 10 U/µL AvaI enzyme, topped up with ultrapure water for a final volume of 20 µL per reaction. The system was maintained at 37 °C for 3 h.

For *IL10* rs1800890, the thermocycling conditions were 94 °C for 5 min (initial denaturation), followed by 35 denaturation cycles at 94 °C for 30 s, oligonucleotide annealing at 62 °C for 45 s, and 72 °C per 55 s for the fragments’ extension. The final extension was performed at 72 °C for 8 min and then cooled for 4 min. Each reaction used 4.0 µL of genomic DNA at the final concentration of 2.5 ng/µL, 2.5 µL of 10× buffer (10 mm of Tris and 50 mm of KCl), 0.5 µL of 50 mM MgCl_2_ (Ludwig Biotec, Alvorada, Rio Grande do Sul, Brazil), 0.5 µL of 2.5 mM deoxyribonucleotide triphosphates (dNTPs) (Ludwig Biotec, Alvorada, Rio Grande do Sul, Brazil), 0.5 µL of 5 U/µL Taq-Polymerase (Ludwig Biotec, Alvorada, Rio Grande do Sul, Brazil), and 1.5 µL of each forward and reverse oligonucleotide (10 µM, IDT Technologies), topping with ultrapure water to a final volume of 25 µL per reaction.

The *IL10* PCR product, a 228pb fragment, was digested using the *Apo*I enzyme (Thermo Fisher Scientific, Waltham, MA, USA). Allele 1 (T) creates a new restriction site, cleaving it into two fragments of 121pb and 107pb, while allele 2 (A) remains uncleaved by the *Apo*I enzyme. Thus, the *IL10* rs1800890 SNP is divided into cleavage genotype (TT), heterozygote (TA), and non-cleavage genotype (AA). This digestion system was assembled using 10.0 μL of PCR product, 5 μL of 10× UB buffer, and 1 μL of 10 U/µL *Apo*I enzyme, completing with ultrapure water for a final volume of 50 μL per reaction. The system was kept at 37 °C for 3 h.

*IL10* and *IL1B* digestion products underwent an electrophoretic run in a 3% agarose gel stained with 0.1% ethidium bromide at 100 W power for 20 min.

### 2.7. Statistical Analysis

For statistical analysis, absolute and relative frequency distribution was applied for categorical variables and quartiles for continuous variables, with continuous data expressed as percentiles. The study utilized Spearman’s rank correlation coefficient to examine the correlation between continuous data on anthropometric, biochemical, and immunological measurements and cytokine dosage levels. For clinical characteristics expressed as categorical data, the non-parametric statistic Kruskall–Wallis H test was used to evaluate the difference between groups’ serum cytokine levels, as the normality assumptions were unmet. The chi-square test of independence was used to determine whether there was an association between categorical variables (MetS presence versus polymorphism). Tests were conducted using SPSS software (version 28.0, SPSS Inc., Chicago, IL, USA) at a 5.0% significance level.

## 3. Results

### 3.1. IL-10 and IL-1β Serum Concentrations’ Associations with Hematological, Biochemical, and Anthropometric Measurements

Correlation analysis in Table 1 revealed that serum IL-10 levels weakly correlated with serum HDL cholesterol levels (ρ = 0.365) and fat mass (ρ = 0.293). Serum IL-10 levels were also inversely associated with serum levels of triglycerides (ρ = −0.268), glucose (ρ = −0.370), glycated hemoglobin (HbA1c) (ρ = −0.379), and estimated average blood glucose (ρ = −0.377). The cytokine was also indirectly associated with three variables: the time with T2DM (ρ = −0.264), the amount of lean mass (ρ = −0.263), and the amount of bone mineral content (ρ = −0.328). On the other hand, serum IL-1β levels correlated directly with two variables: total leukocytes (ρ = 0.213) and the number of segmented neutrophils (ρ = 0.163) (*p* < 0.05; Table 1).

### 3.2. Participants’ Serum IL-10 and IL-1β Levels and Other Clinical Signs and Symptoms

Statistically significant differences were observed between the groups with and without metabolic syndrome (MetS), even as participants with MetS had reduced serum cytokine levels (*p* < 0.001; Table 2), regarding the research participants’ clinical status and serum IL-10 concentrations (Table 2). For instance, participants over 80 exhibited a decrease in IL-10 concentration compared to the rest (*p* < 0.05; Table 2). Similarly, participants with altered HDL levels showed reduced serum IL-10 levels compared to the others (*p* < 0.001; Table 2). Participants with elevated serum glucose and HbA1c levels also exhibited lower IL-10 concentrations than those with normal levels (*p* < 0.001; Table 2). Regarding biological sex, women had higher IL-10 concentrations in their blood compared to men (*p* < 0.001; Table 2). As for serum IL-1β concentrations, the only variable that showed a statistically significant difference was alcohol consumption (Table 2). Surprisingly, participants who consumed alcoholic beverages showed lower variations in serum IL-1β levels compared to those who did not (*p* < 0.05; Table 2).

### 3.3. IL10 and IL1B Gene Polymorphisms’ Genotype Frequency Distribution and Their Relationship with Their Serum Concentrations

This study investigated the *IL10* rs1800890 (-3575T>A) gene variant (Hardy–Weinberg equilibrium (HW): *p* < 0.05) and its effect on serum IL-10 concentrations in older adults (over 60). We found that the different genotypes were associated with significant changes in serum IL-10 levels. Specifically, the presence of the ancestral genotype (TT) (56.5%) correlated with a slight increase in serum IL-10 levels compared to other genotypes (*p* < 0.001).

Similarly, *IL1B* rs1143627 (-31T>C) polymorphism (HW: *p* = 0.1045) also affected serum IL-1β levels. The presence of the polymorphic genotype (TT) (7.8%) correlated with an increase in serum IL-1β concentration compared to other genotypes (*p* < 0.001) (Table 3).

When analyzing the *IL10* -3575T>A (rs1800890) and *IL1Β* -31 T>C (rs1143627) SNPs’ genotypes according to whether the participants had (MetS+) or did not have (MetS−) metabolic syndrome, only *IL10* -3575T>A (rs1800890) presented a statistically significant difference (*p* < 0.01), with the AA genotype being identified only in patients with metabolic syndrome (Figure 1).

## 4. Discussion

Our study’s results analysis revealed a series of correlations between IL-10 and IL-1β serum levels and several clinical variables, as well as with IL10 -3575T>A (rs1800890) and IL1Β -31 T>C (rs1143627) single nucleotide polymorphisms (SNPs) in older adults (over 60) with or without metabolic syndrome. These correlations offer valuable insights into the role of IL-10 and IL-1β in the disease and in the aging process, although they require a cautious approach due to the complexity of the subject.

As people age, the likelihood of developing conditions such as obesity, hypertension, and other chronic noncommunicable diseases (NCDs) increases [11,31]. Our studied population covers a range of age groups over 60, with a higher prevalence of people between 60 and 65. This range allows for a deeper analysis of the aging process and variables analyzed. For example, people above 80 had lower levels of IL-10 than the rest of the groups. Notably, some participants already had chronic health conditions such as type 2 diabetes mellitus (T2DM), systemic arterial hypertension (SAH), fibromyalgia, and other inflammatory diseases. Some even had multiple diseases, which are crucial to explore since the interactions between these conditions can affect the correlations found.

Metabolic syndrome (MetS) encompasses several metabolic dysregulations, including insulin resistance, atherogenic dyslipidemia, central obesity, T2DM, and hypertension [4,6,8,9,12], sharing many of its characteristics with frailty syndrome, also known as geriatric syndrome [32]. MetS development results from a complex interaction between genetic, epigenetic, and environmental factors [6,8,12]. Furthermore, obese women with T2DM showed an increase in the IL-10 expression and secretion in adipose tissue compared to other groups, a difference that has not been observed in men [33]. Nevertheless, another study associated low circulating IL-10 levels with MetS in obese women [15,16]. Our study also noticed a higher IL-10 concentration in women and an inverse correlation with the patients’ time with T2DM (T2DM duration).

Our study reveals positive and inverse correlations between serum IL-10 levels and clinical markers. Specifically, a remarkable positive correlation exists between IL-10, serum HDL levels, and fat mass. Previous studies identified significant associations between IL-10 levels and dyslipidemia; IL-10 levels have been inversely correlated with serum levels of total cholesterol, triglycerides, LDL, and VLDL and positively correlated with HDL levels [34]. Obese children with hypertriglyceridemia also presented decreased IL-10 expression in their serum and adipose tissue [17]. Moraitis et al. [35] observed an inverse relationship between IL-10 and HDL levels, indicating that IL-10 can reduce HDL and LDL cholesterol levels while increasing triglyceride levels. Similarly, Ribeiro et al. [36] found an inverse association between increased fat mass in older women and decreased serum HDL concentrations. The authors also noted that IL-10 strongly influenced body composition, as it correlated positively with waist circumference, the waist-to-hip circumference ratio, and the waist-to-height ratio [36]. These findings highlight the complexity of IL-10’s role in lipid and adipose regulation.

IL-10, despite not directly impacting human adipocyte activity, exerts its influence by regulating lipid metabolism and inflammation in adipose tissue [37]. This indirect effect is crucial, as changes in adipose tissue related to the increased oxidative stress and inflammatory response, likely caused by ROS and proinflammatory adipocytokine production by adipocytes and macrophages, are known to reduce insulin sensitivity [32,38]. Acosta et al. [37] reported that proinflammatory macrophages are the ones that primarily produce IL-10 in obese and insulin-resistant individuals. Additionally, Blüher et al. [39] found that IL-10 levels are lower in individuals with impaired glucose tolerance or T2DM than those with normal glucose tolerance. We also observed that IL-10 was indirectly correlated with triglycerides, glucose, HbA1c, and estimated mean blood glucose. Collectively, these findings suggest that multiple factors may mediate complex associations and merit further investigation.

The statistically significant difference in IL-10 levels between the groups with and without metabolic syndrome suggests a possible association with this condition. However, the clinical relevance of these findings must be carefully considered, as several variables in addition to IL-10 influence metabolic syndrome. Low IL-10 levels are associated with metabolic syndrome, and obesity is correlated with increased levels of this cytokine [15,16]. In a longitudinal study of Iranian adults, a decrease in IL-10 levels was observed in participants with persistent metabolic syndrome [40]. Patients with metabolic syndrome have a distinct inflammatory pattern, as the syndrome affects the expression of proinflammatory genes in adipose tissue and the circulating cytokine levels [41].

Like IL-10’s implications in metabolic syndrome, IL-1β plays a role in this intricate scenario. IL-1β is associated with low-grade chronic inflammatory states, and an increase in its serum concentration often results from chronic diseases, adipose tissue inflammation, and oxidative stress [42]. Although its role, like IL-10, goes further than inflammation into the realms of a critical metabolic regulator [24], we only found its classic role in leukocyte migration.

Our study found direct correlations between IL-1β serum levels and the number of total leukocytes and segmented neutrophils. These correlations confirm IL-1β’s pivotal role in mediating the immune response and systemic inflammation, which frequently involve increased leukocyte and neutrophil counts—particularly in response to chronic inflammation. IL-1β, produced primarily by macrophages but also by other cells, including neutrophils, can induce the expression of other proinflammatory cytokines [43,44]. This expression, in turn, increases leukocyte adhesion to the endothelium and promotes leukocyte chemotaxis, increasing leukocyte and neutrophil counts at sites of inflammation [43,44]. Several other factors can notably influence the relationship between IL-1β and leukocyte and neutrophil counts, including inflammation type and duration, other cytokines and growth factors, and individual patient characteristics [43,45,46].

The discovery of significant differences in serum IL-1β levels based on alcohol consumption is intriguing. It suggests that alcohol consumption may influence a reduction in IL-1β production or release in the body by triggering inflammatory responses and raises questions about the role of alcohol in modulating systemic inflammation in older adults. In the literature, chronic alcohol consumption can have substantial impacts on the immune system, leading to changes in inflammatory cells and triggering dysfunctional immune responses (inability to respond effectively to threats), which, in turn, are associated with various pathologies and an increased susceptibility to infections [47,48,49]. Our findings go against this. However, it is worth remembering that alcohol consumption in our study was self-reported.

Functional gene polymorphisms, such as *IL10* -3575T>A (rs1800890) and *IL1B* -31 T>C (rs1143627) single nucleotide polymorphisms (SNPs), also affect these cytokines’ serum levels [13,27,28,29]. Our study likewise associated different *IL10* -3575T>A (rs1800890) and *IL1B* -31 T>C (rs1143627) SNP genotypes with their serum levels, confirming the evidence of these cytokines’ genetic regulation. We found that the IL*10* ancestral genotype (TT) correlated with IL-10 higher levels. At the same time, the *IL1B* mutated genotype (TT) correlated with higher IL-1β concentrations. However, only the *IL10* -3575T>A (rs1800890) SNP genotypes correlated significantly with metabolic syndrome, with the AA genotype having a higher frequency of participants with metabolic syndrome. This correlation makes sense as our study with older adults (over 60) found that higher IL-10 levels were inversely associated with many MetS clinical characteristics, as discussed above. In contrast, IL-1β levels in our study correlated directly with total leukocytes, the number of segmented neutrophils, and self-reported alcohol consumption. These differences in genotypes underscore the role of genetics in the inflammatory response, as specific genotypes may be associated with distinct immune response profiles, thus highlighting the significance of tailoring treatment and therapeutic approaches to a patient’s genetic characteristics.

However, any interpretation must consider Brazilian demographic diversity, the presence of single or multiple chronic diseases, interactions between variables, and the patient’s genetic characteristics and environmental contexts. Causality cannot be inferred from our correlations as other unconsidered or uncontrolled factors may influence the observed associations; for instance, some of the parameters were self-reported. Nevertheless, our findings are crucial for comprehending the mechanisms that underlie health conditions in older adults and can direct future research and more targeted therapeutic approaches.

## 5. Conclusions

This study underscores the intricate interplay between serum IL-10 and IL-1β levels and numerous variables in an older population over 60. It validates the influence of the *IL10* rs1800890 (-3575T>A) and *IL1B* rs1143627 (-31T>C) gene variants on interleukin concentrations, with specific genotypes correlating with increased levels of respective interleukins and only the *IL10* -3575T>A (rs1800890) variant correlating significantly with metabolic syndrome. While IL-10 levels correlated weakly with HDL cholesterol, fat mass, and several metabolic markers, IL-1β levels were directly associated with leukocyte counts and segmented neutrophils. Certain demographic factors such as age and gender and health conditions like altered lipid and glucose levels impacted interleukin levels. However, caution is warranted in interpreting causality due to the complex interplay of unconsidered and uncontrolled factors. It is also essential to recognize that participants self-declared some evaluated parameters, such as alcohol consumption, which may introduce bias or inaccuracy in the results. Nonetheless, these findings provide valuable insights into the underlying mechanisms of health conditions in older adults, guiding future research and therapeutic strategies.

## Figures and Tables

**Figure 1 nutrients-16-01241-f001:**
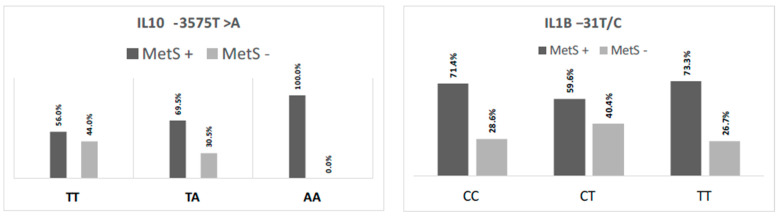
*IL10* -3575T>A (rs1800890) and *IL1Β* -31 T>C (rs1143627) single nucleotide polymorphisms’ genotype distribution, according to the participants’ metabolic syndrome (MetS+ or −) profile.

**Table 1 nutrients-16-01241-t001:** Correlations between IL-10 and IL-1β serum levels with biochemical, immunological, and anthropometric parameters.

Parameters		[*IL-10*] pg/mL	[IL-1 β] pg/mL
[*IL-10*] pg/mL	ρ	1.000	0.016
	*p*-value	-	0.825
[IL-1 β] pg/mL	ρ	0.016	1.000
	*p*-value	0.825	-
Total leukocytes	ρ	−0.06	0.213 **
	*p*-value	0.406	0.003
Rod neutrophils	ρ	0.011	0.093
	*p*-value	0.878	0.201
Segmented neutrophils	ρ	−0.02	0.163 *
	*p*-value	0.781	0.024
Eosinophils	ρ	−0.096	0.046
	*p*-value	0.187	0.530
Basophils	ρ	−0.104	0.028
	*p*-value	0.154	0.703
Lymphocytes	ρ	0.053	−0.131
	*p*-value	0.467	0.070
Monocytes	ρ	−0.068	−0.104
	*p*-value	0.350	0.153
Total cholesterol	ρ	0.008	0.050
	*p*-value	0.910	0.492
Triglycerides	ρ	−0.268 **	0.079
	*p*-value	<0.001	0.272
HDL	ρ	0.365 **	0.038
	*p*-value	<0.001	0.598
LDL	ρ	0.003	0.034
	*p*-value	0.972	0.648
Total lipids	ρ	−0.023	0.106
	*p*-value	0.830	0.311
Glucose	ρ	−0.370 **	0.017
	*p*-value	<0.001	0.817
HbA1c	ρ	−0.379 **	0.052
	*p*-value	<0.001	0.476
Estimated average blood glucose	ρ	−0.377 **	0.056
	*p*-value	<0.001	0.442
Glutamic oxaloacetic transaminase	ρ	0.115	−0.010
	*p*-value	0.111	0.894
Pyruvic glutamic transaminase	ρ	0.016	−0.113
	*p*-value	0.821	0.116
Time with T2DM	ρ	−0.264 **	−0.031
	*p*-value	<0.001	0.677
Time with SAH	ρ	−0.071	−0.053
	*p*-value	0.332	0.471
BMI	ρ	0.109	0.038
	*p*-value	0.146	0.608
Waist circumference	ρ	−0.150	0.073
	*p*-value	0.062	0.366
Fat mass	ρ	0.293 **	0.011
	*p*-value	<0.001	0.896
Lean Mass	ρ	−0.263 **	−0.03
	*p*-value	<0.001	0.707
Bone mineral content	ρ	−0.328 **	−0.012
	*p*-value	<0.001	0.879

**Caption:** ρ: Spearman’s rank correlation coefficient; *p*-value: Mann–Whitney U test; IL-10—interleukin 10; IL-1β—interleukin 1 beta; *—*p* < 0.05; **—*p* < 0.001; <—less than; SAH—systemic arterial hypertension; T2DM—type 2 diabetes mellitus; BMI—body mass index; HDL—high-density lipoprotein; HbA1c—glycated hemoglobin; LDL—low-density lipoprotein.

**Table 2 nutrients-16-01241-t002:** Participants’ serum IL-10 and IL-1β concentration levels according to the presence/absence of clinical characteristics, signs, symptoms, or lifestyle habits.

		[*IL-10*] pg/mL					[*IL-1β*] pg/mL				
		N	P25	Median	P75	*p* Value	N	P25	Median	P75	*p* Value
MetS	Yes	127	4.28	4.59	4.84	<0.001 *	127	3.53	6.23	8.66	0.482
	No	66	5.24	5.35	5.57		66	4.74	7.47	8.75	
Sarcopenia (DEXA)	Yes	23	4.25	4.84	5.28	0.722	23	4.83	7.44	8.75	0.537
	No	170	4.51	4.85	5.23		170	3.53	7.30	8.75	
SAH	Yes	147	4.35	4.84	5.22	0.266	147	3.53	6.23	8.75	0.231
	No	46	4.37	5.09	5.34		46	4.77	7.48	8.78	
T2DM	Yes	108	4.34	4.83	5.12	0.007 *	108	3.53	7.30	8.68	0.821
	No	85	4.51	5.02	5.38		85	3.54	7.40	8.76	
Age	60 to 65	81	4.51	4.84a	5.29	0.042 *	81	3.54	6.25	7.53	0.691
	66 to 69	43	4.51	4.83a	5.06		43	3.53	7.47	8.84	
	70 to 75	40	4.35	5.045a	5.31		40	3.52	6.02	8.76	
	76 to 79	16	4.67	5.035a	5.27		16	4.86	7.51	8.81	
	≥80	13	4.11	4.48b	4.84		13	3.65	7.53	8.66	
Smoking	Yes	14	4.31	4.94	5.24	0.808	14	3.50	6.85	8.75	0.644
	No	179	4.37	4.84	5.24		179	3.54	7.40	8.75	
Use of alcoholic beverages	Yes	9	4.48	5.01	5.24	0.709	9	3.51	3.53	5.94	0.037 *
	No	184	4.37	4.84	5.24		184	3.55	7.42	8.75	
Perform physical exercises	Yes	135	4.37	4.86	5.27	0.573	135	3.54	7.30	8.75	0.853
	No	58	4.48	4.84	5.22		58	3.53	7.40	8.76	
Altered BP	Yes	80	4.35	4.84	5.22	0.358	80	3.52	7.41	8.77	0.596
	No	95	4.51	4.86	5.24		95	4.74	7.48	8.76	
Altered SBP	Yes	56	4.33	4.79	5.16	0.241	56	3.52	6.24	8.73	0.288
	No	116	4.51	4.86	5.24		116	4.71	7.48	8.77	
Altered DBP	No	116	4.37	4.86	5.23	0.954	116	4.71	7.48	8.75	0.301
	Yes	53	4.48	4.83	5.22		53	3.52	6.25	8.79	
HDL	Altered	40	4.24	4.51	4.93	<0.001 *	40	3.53	7.46	8.81	0.709
	Normal	152	4.59	5.01	5.28		152	3.54	7.30	8.75	
Glucose	Altered	98	4.29	4.76	5.01	<0.001 *	98	3.53	6.27	7.65	0.567
	Normal	95	4.79	5.22	5.40		95	4.68	7.41	8.76	
HbA1c	Altered	110	4.29	4.77	5.02	<0.001 *	110	3.54	7.46	8.75	0.341
	Normal	82	4.79	5.19	5.38		82	3.52	6.19	8.69	
Total cholesterol	Altered	102	4.48	4.86	5.24	0.806	102	3.54	7.41	8.76	0.911
	Normal	91	4.35	4.84	5.22		91	3.53	6.29	8.75	
LDL	Altered	100	4.37	4.84	5.24	0.945	100	3.54	7.46	8.79	0.891
	Normal	87	4.35	4.84	5.22		87	3.53	6.29	8.75	
Waist circumference	Altered	123	4.51	4.84	5.07	0.851	123	3.53	7.30	8.75	0.343
	Normal	32	4.28	4.82	5.30		32	3.52	6.23	7.50	
Biological sex	Female	154	4.74	5.01	5.28	<0.001 *	154	3.54	7.41	8.75	0.588
	Male	39	4.25	4.31	5.18		39	3.53	6.09	8.79	

**Caption:** IL-10—interleukin 10; IL-1β—interleukin 1 beta; N—number of participants; *—*p* < 0.05; <—less than; MetS—metabolic syndrome; P25—25th percentile; P75—75th percentile; DEXA—dual-energy X-ray absorptiometry; SAH—systemic arterial hypertension; T2DM—type 2 diabetes mellitus; BP—blood pressure; SBP—systolic blood pressure; DBP—diastolic blood pressure; HDL—high-density lipoprotein; HbA1c—glycated hemoglobin; LDL—low-density lipoprotein. **Note:** Different letters denote statistical differences in different groups.

**Table 3 nutrients-16-01241-t003:** *IL10* -3575T>A (rs1800890) and *IL1Β* -31 T>C (rs1143627) single nucleotide polymorphisms’ genotype distribution, according to the participants’ profile.

	IL10 -3575T>A			IL1Β −31T>C		
	TT	TA	AA	CC	CT	TT
	[*IL-10*] pg/mL	[*IL-10*] pg/mL	[*IL-10*] pg/mL	[*IL-1Β*] pg/mL	[*IL-1Β*] pg/mL	[*IL-1Β*] pg/mL
N	109	59	25	84	94	15
%	56.5	30.6	12.9	43.5	48.7	7.8
P25	4.76	4.28	4.02	3.49	7.48	17.46
Median	5.02a	4.84b	4.23c	3.52a	7.53b	17.52c
P75	5.34	5.23	4.29	4.76	8.82	18.66
*p*-value		<0.001 *			<0.001 *	

**Caption:** IL10—interleukin 10 gene; IL1Β—interleukin 1 beta gene; N—number of participants; %—percentage; <—less than; *—*p* < 0.05; P25—25th percentile; P75—75th percentile. Note: Different letters denote statistical differences in different groups (specific genotypes) in relation to IL-10 and IL-1β levels.

## Data Availability

The research data are contained in the article’s tables.

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
