# Peer review of "IL-10 and IL-1β Serum Levels, Genetic Variants, and Metabolic Syndrome: Insights into Older Adults’ Clinical Characteristics"

_nutrients, 2024, doi:10.3390/nu16081241_

Round 1
Reviewer 1 Report
Comments and Suggestions for Authors
Renata de Souza Freitas et al., in " IL-10 and IL-1β Serum Levels, Genetic Variants, and Metabolic 2 Syndrome: Insights into Older Adults' Clinical Characteristics" provide a comparative study concerning genotype frequencies of IL1B rs1143627 (–31T>C) and IL10 rs1800890 (-3575T>A) variants in a population over 60 to their respective serum concentrations.
They outline that those with the IL10 AA genotype (lower IL-10 levels) had a significantly higher risk of developing MetS.
Despite the potential interesting topic, the paper suffers from some issues related to the methodology used in data analysis and discussion of the results.
Here are some comments to consider in order to improve the content of this manuscript:
1. In the Results section, authors stated that “The cytokine was also indirectly influenced by three variables: the time with T2DM, the amount of lean mass, and the amount of bone mineral content (please see lines 217-218); we consider that correlational analysis (presented in Table 1) does not test the influence or directionality of the association ⇒ please reformulate the statement
2. Authors stated that “Statistically significant differences were observed between the groups with and without metabolic syndrome (MetS) regarding the research participants' clinical status and serum IL-10 concentrations (serum cytokine levels were reduced in participants with MetS)” ; The statement does not refer to the table or any other statistical result (besides, from the table, only the association with IL-10 is observed)
3. Authors stated that “For instance, participants over 80 exhibited a decrease in IL-10 concentration compared to the rest” but where is the numerical result??
4. The authors should verify if the genotypes of studied genetic polymorphisms showed no deviations from Hardy-Weinberg Equilibrium.
5. In the legend of Table 3 authors mentioned that “Note: Different letters denote statistical differences in different groups”, please specify between which groups significant differences were identified.
6. The association analysis between IL-10 and IL-1β should be adjusted for potential confounding factors (age, sex, BMI, smoking, alcohol consumption)
7. In the discussion section, authors stated that “Our study found direct correlations between IL-1β serum levels and the number of total leukocytes and segmented neutrophils” but the correlational analysis cannot study the direct or indirect role of cytokines, but possibly a mediational analysis
8. Authors stated that “However, only the IL10 -3575T>A (rs1800890) SNP genotypes correlated significantly with metabolic syndrome, with the AA genotype conferring a higher risk of having metabolic syndrome.” but we found no result in the Results section to confirm this statement, only a p-value from a chi-squared test obtained for the entire distribution of genotypes.
Author Response
We appreciate the suggestions and have addressed each point raised by the reviewers. We believe these suggestions have improved the overall quality of the submitted manuscript.
Therefore, we are resubmitting our revised systematic review entitled "IL-10 and IL-1β Serum Levels, Genetic Variants, and Metabolic Syndrome: Insights into Older Adults' Clinical Characteristics". Changes in the manuscript and our responses to the reviewers' comments are in blue.
All authors are aware of the resubmission and agree with the following responses to the reviewers.
__________________________________________________
Reviewer 1
Comments and Suggestions for Authors
- In the Results section, authors stated that “The cytokine was also indirectly influenced by three variables: the time with T2DM, the amount of lean mass, and the amount of bone mineral content (please see lines 217-218); we consider that correlational analysis (presented in Table 1) does not test the influence or directionality of the association ⇒ please reformulate the statement.
Answer: We agree and rephrased the sentence to: "The cytokine was also indirectly associated with three variables: the time with T2DM, the amount of lean mass, and the amount of bone mineral content.”
- Authors stated that “Statistically significant differences were observed between the groups with and without metabolic syndrome (MetS) regarding the research participants' clinical status and serum IL-10 concentrations (serum cytokine levels were reduced in participants with MetS)” ; The statement does not refer to the table or any other statistical result (besides, from the table, only the association with IL-10 is observed)
Answer: This statement was an introductory (catch) phrase for Table 2’s results. We have added ‘(Table 2)’ at the end of the clause.’ We also corrected the sentence to:
‘Statistically significant differences were observed between the groups with and without metabolic syndrome (MetS), even as participants with MetS had reduced serum cytokine levels (P<0.001; Table 2), regarding the research participants' clinical status and serum IL-10 concentrations (Table 2).’
- Authors stated that “For instance, participants over 80 exhibited a decrease in IL-10 concentration compared to the rest” but where is the numerical result??
Answer: We added ‘(P< 0.05; Table 2)’ at the end of the phrase. We also similarly added (P<X; Table 2) for the other phrases.
- The authors should verify if the genotypes of studied genetic polymorphisms showed no deviations from Hardy-Weinberg Equilibrium.
Answer: Thank you for pointing out the absence of this information. For our sample, the Hardy-Weinberg equilibrium (HW) was p= 0.1045 for the IL1Β −31T>C variant and P<0.05 for the IL10 -3575T>A variant. We've added this information to our manuscript:
‘This study investigated the IL10 rs1800890 (-3575T>A) gene variant [Hardy-Weinberg equilibrium (HW): p<0.05] and its effect on serum IL-10 concentrations in older adults (over 60). We found that the different genotypes were associated with significant changes in serum IL-10 levels. Specifically, the presence of the ancestral genotype (TT) (56.5%) correlated with a slight increase in serum IL-10 levels compared to other genotypes (p<0.001).
Similarly, IL-1β rs1143627 (–31T>C) polymorphism [HW: p=0.1045] also affected serum IL-1β levels. The presence of the Polymorphic Genotype (TT) (7.8%) correlated with an increase in serum IL-1β concentration compared to other genotypes (p<0.001) (Table 3). ‘
Our sample, in which we analyzed both gene variants, is unique in its composition. The participants were specifically chosen from patients treated at a Basic Health Unit (UBS) in the Ceilândia Administrative Region of the Federal District - Brazil between February and June 2019, based on their adherence to our inclusion/exclusion criteria. This sample is considered part of a population subdivision or demes rather than the general population and, therefore, might be potentially under a Wahlund effect.
Garnier-Géré, Pauline, and Lounès Chikhi. "Population subdivision, Hardy–Weinberg equilibrium and the Wahlund effect." Els (2013). Doi: https://doi.org/10.1002/9780470015902.a0005446.pub3
- In the legend of Table 3 authors mentioned that “Note: Different letters denote statistical differences in different groups”, please specify between which groups significant differences were identified.
Answer: Different letters denote statistical differences between specific genotypes in relation to IL-10 and IL-1β levels. We modified Table 3's caption to reflect this information:
‘Note: Different letters denote statistical differences in different groups (specific genotypes) in relation to IL-10 and IL-1β levels.’
- The association analysis between IL-10 and IL-1β should be adjusted for potential confounding factors (age, sex, BMI, smoking, alcohol consumption)
Answer: Cytokine level analysis is complex. We've carefully restricted our sample through our inclusion/exclusion criteria to reduce the risk of confounding to better correlate IL-10 and IL-1β serum levels with the participants' biochemical, immunological, and anthropometric parameters. We then analyzed our total sample by different subgroups (strata), such as age or biological sex, analyzing the results separately for each subgroup (Table 2).
- In the discussion section, authors stated that “Our study found direct correlations between IL-1β serum levels and the number of total leukocytes and segmented neutrophils” but the correlational analysis cannot study the direct or indirect role of cytokines, but possibly a mediational analysis
Answer: Spearman's rank correlation coefficient uses a monotonic function to assess the strength and direction of the relationship between two variables. A direct correlation is another term for positive correlation, which means that if one increases, the other does too, and vice versa. Our study found direct (positive) correlations between IL-1β serum levels and the number of total leukocytes and segmented neutrophils, which is as expected given IL-1β chemotactic roles.
- Authors stated that “However, only the IL10 -3575T>A (rs1800890) SNP genotypes correlated significantly with metabolic syndrome, with the AA genotype conferring a higher risk of having metabolic syndrome.” but we found no result in the Results section to confirm this statement, only a p-value from a chi-squared test obtained for the entire distribution of genotypes.
Answer: Figure 1 presents the genotype distribution of the IL10 -3575T>A (rs1800890) and IL1Β -31 T>C (rs1143627) single nucleotide polymorphisms according to the participants' metabolic syndrome profile. In this figure, only the IL10 -3575T>A (rs1800890) AA genotype conferred a higher risk of having metabolic syndrome, with 100% of the participants with this genotype having metabolic syndrome.
We rephrased the sentence to “However, only the IL10 -3575T>A (rs1800890) SNP genotypes correlated significantly with metabolic syndrome, with the AA genotype having a higher frequency of participants with metabolic syndrome”.
Reviewer 2 Report
Comments and Suggestions for Authors
Title, abstract and key worlds meet whole artilce
MAterial and methods
1. Authors used deffinition of Metabolic Syndrome properly, however the Blood Pressure was described as (BP) ≥130 x 85 mmHg it should be changed into 130/85
2. I suggest to move section 2.6. Laboratory Analysis and Cytokine Dosage before genetics description.
3. What is more as main subject is about Interleukins I suspect to extend this part of methods - extend and describe The sensitivity and the intraassay coefficient of variation (CV), i.e., repeatability, for these ELISA tests (as they were done)
The results are clearly described, however I found transaminases in the results - why authors use dhem. What is more the methods did not includ them.
Disscusion must be extended, you may use https://doi.org/10.3390/metabo13040475
Conclusions are too long it look rather like summary of the results.
I also expect descryption of the limitations and strength of the study.
At the end please update the references list - some articles are rather old
Comments on the Quality of English LanguageMinor editing of English language required
Author Response
We appreciate the suggestions and have addressed each point raised by the reviewers. We believe these suggestions have improved the overall quality of the submitted manuscript.
Therefore, we are resubmitting our revised systematic review entitled "IL-10 and IL-1β Serum Levels, Genetic Variants, and Metabolic Syndrome: Insights into Older Adults' Clinical Characteristics". Changes in the manuscript and our responses to the reviewers' comments are in blue.
All authors are aware of the resubmission and agree with the following responses to the reviewers.
__________________________________________________
Reviewer 2
Comments and Suggestions for Authors
Methods
- Authors used definition of Metabolic Syndrome properly, however the Blood Pressure was described as (BP) ≥130 x 85 mmHg it should be changed into 130/85
Answer: We agree with the suggestion and added ‘130/85’ to the text,’ as seen here ‘blood pressure (BP) ≥130 x 85 mmHg (≥ 130/85)’.
- I suggest to move section 2.6. Laboratory Analysis and Cytokine Dosage before genetics description.
Answer: We agree with the suggestion and reorganized the text.
- What is more as main subject is about Interleukins I suspect to extend this part of methods - extend and describe The sensitivity and the intraassay coefficient of variation (CV), i.e., repeatability, for these ELISA tests (as they were done)
Answer: The participants' samples were also done in triplicate, and we added ‘The samples mean values were calculated from these triplicate determinations’ to the methods to facilitate understanding.
The results are clearly described, however I found transaminases in the results - why authors use dhem. What is more the methods did not include them.
Answer: Biochemical and hematological tests, including transaminase measurement, were performed in a private laboratory, as described in section 2.4. Laboratory Analysis and Cytokine Dosage. We included which the tests performed in this laboratory to ensure transparency as follows:
'Biochemical and Hematological tests were performed at a Federal District's private clinical analysis laboratory funded by the research project budget. These exams included the analysis of total leukocytes, rod neutrophils, segmented neutrophils, eosinophils, basophils, lymphocytes, monocytes, and measurements of total cholesterol, triglycerides, HDL, LDL, total lipids, glucose, glycated hemoglobin (HbA1c), estimated mean blood glucose, glutamic oxaloacetic transaminase and glutamic pyruvic transaminase.'
Disscusion must be extended, you may use https://doi.org/10.3390/metabo13040475
Answer: We added https://doi.org/10.3390/metabo13040475 to supplement the discussion.
Conclusions are too long it look rather like summary of the results.
Answer: We agree and summarized the conclusion in text as follows:
“The study underscores the intricate interplay between serum IL-10 and IL-1β levels and numerous variables in an older population over 60. It validates the influence of IL10 rs1800890 (-3575T>A) and IL1B rs1143627 (–31T>C) gene variants on interleukin concentrations, with specific genotypes correlating with increased levels of respective interleukins and only the IL10 -3575T>A (rs1800890) variant correlating significantly with metabolic syndrome. While IL-10 levels correlated weakly with HDL cholesterol, fat mass, and several metabolic markers, IL-1β levels were directly associated with leukocyte counts and segmented neutrophils. Certain demographic factors such as age and gender and health conditions like altered lipid and glucose levels impacted interleukin levels. However, caution is warranted in interpreting causality due to the complex interplay of unconsidered and uncontrolled factors. It is also essential to recognize that participants self-declared some evaluated parameters, such as alcohol consumption, which may introduce bias or inaccuracy in the results. Nonetheless, these findings provide valuable insights into the underlying mechanisms of health conditions in older adults, guiding future research and therapeutic strategies.”
I also expect descryption of the limitations and strength of the study.
Answer: We added a last paragraph in the discussion (as seen below) and altered the conclusion (as seen in the answer above) to reflect this information.
“However, any interpretation must consider Brazilian demographic diversity, the presence of single or multiple chronic diseases, interactions between variables, and the patient's genetic characteristics and environmental contexts. Causality cannot be inferred from our correlations as other unconsidered or uncontrolled factors may influence the observed associations; for instance, some of the parameters were self-reported. Nevertheless, our findings are crucial for comprehending the mechanisms that underlie health conditions in older adults and can direct future research and more targeted therapeutic approaches.”
At the end please update the references list - some articles are rather old
Answer: We revised and updated the references as suggested, replacing some older articles with newer references. Nevertheless, we kept some articles due to their relevance and significant contribution to the topic covered.
Round 2
Reviewer 1 Report
Comments and Suggestions for Authors
The authors attempted to satisfactorily address the requirements.